# Single-cell transcriptomics reveals expression profiles of *Trypanosoma brucei* sexual stages

**Virginia M. Howick**[1,2,3]*, **Lori Peacock**[4,5], **Chris Kay**[4], **Clare Collett**[4¤], **Wendy Gibson**[4☉], **Mara K. N. Lawniczak**[3☉]

**1** Institute of Biodiversity, Animal Health, and Comparative Medicine, University of Glasgow, Glasgow, United Kingdom, **2** Wellcome Centre for Integrative Parasitology, University of Glasgow, Glasgow, United Kingdom, **3** Parasites and Microbes Programme, Wellcome Sanger Institute, Hinxton, United Kingdom, **4** School of Biological Sciences, University of Bristol, Bristol, United Kingdom, **5** Bristol Veterinary School, University of Bristol, Langford, United Kingdom

☉ These authors contributed equally to this work.
¤ Current address: UK Health Security Agency, Porton Down, United Kingdom
* Virginia.Howick@glasgow.ac.uk

**Data Availability Statement:** scRNAseq sequence data are available on the European Nucleotide Archive under accession ERP132258. The expression matrix and associated code are available on Zenodo https://www.doi.org/10.5281/

## Abstract

Early diverging lineages such as trypanosomes can provide clues to the evolution of sexual reproduction in eukaryotes. In *Trypanosoma brucei*, the pathogen that causes Human African Trypanosomiasis, sexual reproduction occurs in the salivary glands of the insect host, but analysis of the molecular signatures that define these sexual forms is complicated because they mingle with more numerous, mitotically-dividing developmental stages. We used single-cell RNA-sequencing (scRNAseq) to profile 388 individual trypanosomes from midgut, proventriculus, and salivary glands of infected tsetse flies allowing us to identify tissue-specific cell types. Further investigation of salivary gland parasite transcriptomes revealed fine-scale changes in gene expression over a developmental progression from putative sexual forms through metacyclics expressing variant surface glycoprotein genes. The cluster of cells potentially containing sexual forms was characterized by high level transcription of the gamete fusion protein HAP2, together with an array of surface proteins and several genes of unknown function. We linked these expression patterns to distinct morphological forms using immunofluorescence assays and reporter gene expression to demonstrate that the kinetoplastid-conserved gene Tb927.10.12080 is exclusively expressed at high levels by meiotic intermediates and gametes. Further experiments are required to establish whether this protein, currently of unknown function, plays a role in gamete formation and/or fusion.

## Author summary

African Trypanosomes are single-celled protozoan parasites that cause disease in humans and livestock. They have a complex life cycle that spans a mammalian and tsetse fly host. Within the tsetse fly, the parasite first travels into the midgut when the fly takes up an infectious blood meal. As it develops it moves into the proventriculus followed by the salivary glands taking on distinct morphological forms in each of these tissues. In the salivary

zenodo.6047732. The data are explorable via the Glasgow Cell Atlas website at http://cellatlas.mvls.gla.ac.uk/.

**Funding:** We are grateful to the UK Biotechnology and Biological Sciences Research Council (https://bbsrc.ukri.org/), Wellcome Trust (http://www.wellcome.ac.uk/) and Royal Society (https://royalsociety.org) for funding. This research was supported by BBSRC Grants BB/R016437/1 and BB/R010188/1 to WG, Wellcome Trust Grant 206194/Z/17/Z to the Wellcome Sanger Institute, and a Sir Henry Dale Fellowship jointly funded by the Wellcome Trust and the Royal Society (Grant 220185/Z/20/Z) to VMH. LP, CK and CC received salary from BBSRC. VMH and MKNL received salary from Wellcome Trust. The funders had no role in study design, data collection and analysis, decision to publish, or preparation of the manuscript.

**Competing interests:** The authors have declared that no competing interests exist.

glands, the parasite can undergo non-obligatory sexual reproduction via meiosis and the production of gametes. However, the biological processes that underly this sexual developmental and the molecular signatures that define these morphological forms remain elusive because they are found within heterogeneous populations that also contain mitotically dividing forms. Here we have used single-cell RNAseq to profile the transcriptomes of parasites across development in the tsetse with a focus on identifying the patterns of expression that define these sexual stages. We showed that the sexual forms have a unique transcriptional profile and we connect these expression patterns to specific morphological stages of sexual development using a fluorescent reporter. This allowed us to identify a new gene that may be involved in reproduction. Elucidating the mechanisms underlying sexual reproduction and genetic exchange is fundamental to understanding the evolution of key traits such as virulence and drug resistance.

## Introduction

The African tsetse-transmitted trypanosomes are single-celled parasites that cause human and animal diseases, which are a heavy burden for many countries in sub-Saharan Africa. These trypanosomes survive in both the tsetse and mammalian host by taking on distinct morphological forms that suit the diverse metabolic and immune environments they encounter [1]. When blood infected with *Trypanosoma brucei* is imbibed by the tsetse fly (genus *Glossina*), trypanosome blood stream forms (BSF) rapidly change their transcriptional profile, including switching off Variant Surface Glycoprotein (VSG) transcription and upregulating expression of other surface proteins such as procyclins [2,3] They also switch their metabolism from dependence on glucose processed via glycolysis in the glycosome to exploitation of amino acids such as proline via the mitochondrial TCA cycle [4]. Trypanosomes then multiply as procyclics in the fly midgut before migrating anteriorly, first colonising the proventriculus or cardia, the valve between the foregut and anterior midgut, and then the paired salivary glands (**Fig 1A**) [5,6]. Here trypanosomes attach and proliferate as epimastigotes characterised by BARP surface proteins [7], before final differentiation into infective metacyclics that are inoculated into a new host via the saliva.

Additionally, the salivary glands are the location of the non-obligatory sexual cycle of *T. brucei*, which involves meiosis and the production of haploid gametes [8,9]. As trypanosomes are early diverging eukaryotes, their sexual processes are of particular interest because they provide insights into the evolution of sexual reproduction and meiosis. Although the morphologies of the meiotic division stages and gametes have been described [8–10], little is known about the transcriptional dynamics that characterise the sexual stages because these cells are a minority of the heterogenous cell population in the salivary glands. Sexual stages are found during the early phase of establishment of salivary gland infection, with numbers peaking about three weeks after fly infection [8,9]. The sexual cycle appears to be a sideshow in the normal mitotic developmental program, as it occurs in clonal trypanosome lines and does not need to be triggered by external factors such as the presence of another strain.

Single-cell RNAseq opens the door to study heterogenous populations of single-celled parasites by delineating expression patterns of individual cells allowing us to understand continuous developmental processes, cell-type specific patterns of co-expression and bet-hedging strategies [11–17]. Recent studies have profiled *T. brucei* populations using single-cell droplet-based approaches from BSF culture to profile the development of stumpy forms [18] as well as *in vivo* salivary gland parasites in order to identify a potential vaccine candidate among mature

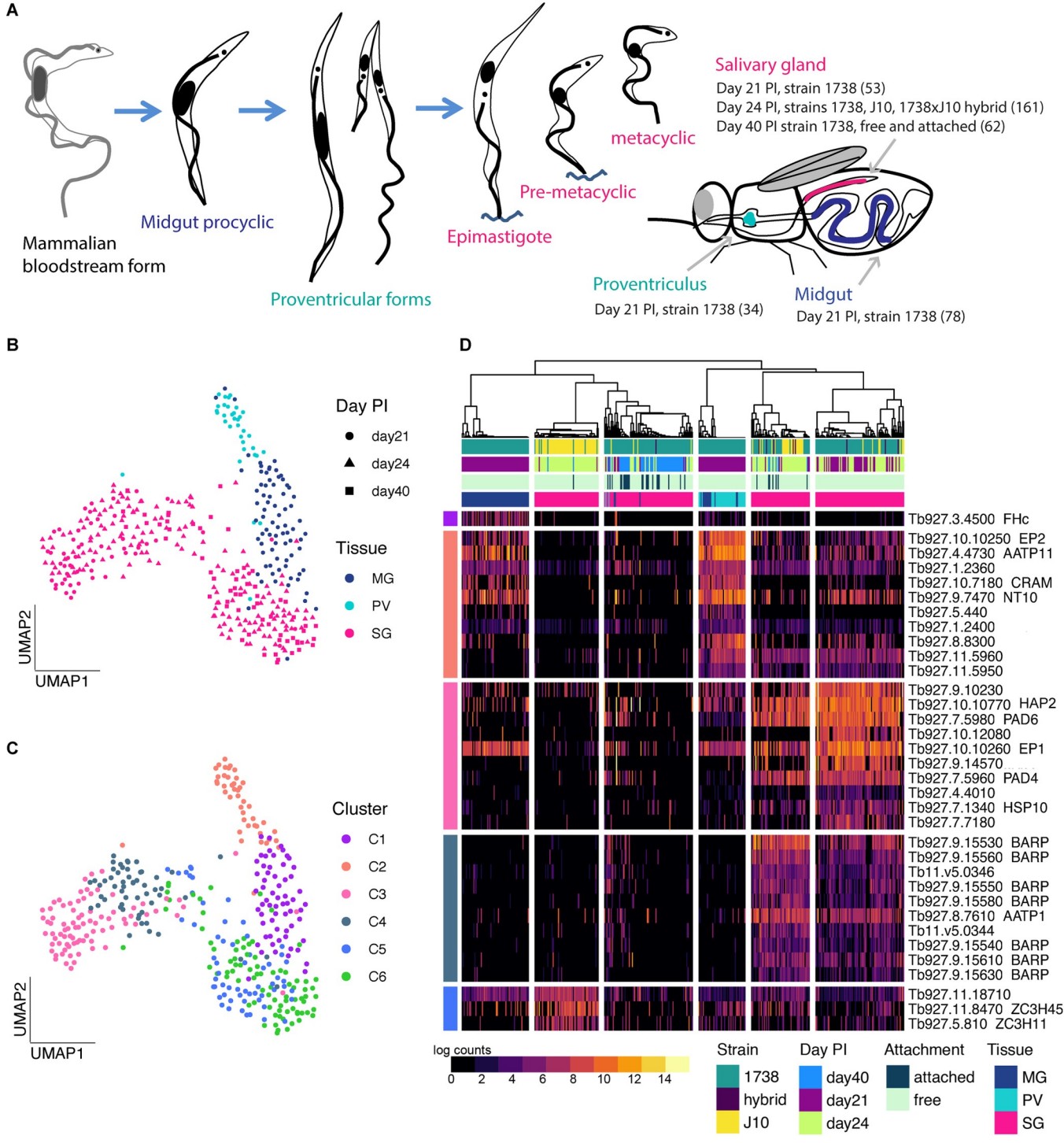

**Fig 1. scRNAseq analysis of trypanosome developmental stages in tsetse.** (A) A schematic of the trypanosome life cycle and collections of the single parasite transcriptomes from midgut (MG; blue), proventriculus (PV; turquoise) and salivary glands (SG; pink) from different time points and strains. The number of parasites that passed QC at each collection is shown in parentheses. Trypanosomes show two conformations: trypomastigote with kinetoplast (small black dot) posterior to nucleus (*e.g.* bloodstream form, procyclic, metacyclic) and with kinetoplast anterior to nucleus (*e.g.* epimastigote). (B) A UMAP of the 388 cells that passed QC across collections, coloured by tissue of origin. (C) The UMAP coloured by cluster assignment. (D) A heatmap of the top significant marker genes from each of the five clusters that had marker genes (AUROC >0.75 & adjusted *p*-value < 0.01).

metacyclics [19] and the dynamics of *VSG* expression in the developing metacyclic parasites [20]. These studies complement the previous bulk transcriptomic studies in *T. brucei* that identified the major changes in transcriptional patterns over time and metacyclic development using either whole infected salivary glands or *in vitro*-derived metacyclics from the RBP6-inducible system [21–28]. Although these studies have been essential to our understanding of the dynamics of gene expression in *T. brucei*, we still lack an understanding of the molecular processing that characterize meiosis and sexual development in kinetoplastids.

Here we have exploited scRNAseq to investigate transcriptomes of the sexual stages of *T. brucei* that occur transiently in the heterogeneous trypanosome population in the fly salivary glands. We used a modified Smart-seq2 protocol [12,13,29] to profile *T. brucei* cells from different tsetse tissues (midgut, proventriculus, salivary glands) at different time points during development. We tied the observed transcriptomic profiles to specific developmental stages, validated by immunofluorescence, and identified cell-type specific markers, which revealed the dynamics of surface protein expression as well as a new candidate gene that may be involved in sexual development.

## Results and discussion

### Generation of high-quality transcriptomes from *in vitro* procyclic forms

To confirm that the modified Smart-seq2 protocol produces high-quality data for *T. brucei*, we initially profiled 46 single-cell transcriptomes from *in vitro* procyclic forms. We found a mean of $2.6 \times 10^{6}$ mapped reads per cell and a mean detection of 1756 genes per cell (**S1A and S1B Fig**), which is a greater number of genes per cell than recently published data from Hutchinson et al 2021 [20] that used a droplet-based method on the same parasite stage (1258 genes per cell). This further supports the use of Smart-seq2 to get in depth transcriptomes (high-coverage and full-length) in a low-throughput, targeted fashion compared to droplet-based methods that have fewer genes detected but are higher throughput [12,30]. Additionally, we observed high expression of genes that encode known procyclic surface antigens including GPEET and EP1-3 [2,31] (**S1C Fig**). These data support the utility of our protocol to profile single-cell expression profiles in kinetoplastids.

### Transcriptomes of fly developmental stages

Having confirmed that the modified Smart-seq2 protocol would produce high-quality data from *T. brucei in vitro* procyclic forms, we profiled parasite transcriptomes isolated from diverse tsetse tissues at different timepoints, and from two *T. brucei* strains as outlined in **Fig 1A**. After quality control (**S2 Fig**), we obtained a total of 388 single-cell parasite transcriptomes: 78 from the midgut, 34 from the proventriculus and 276 from the salivary glands (**Fig 1A**). Parasites from all three tissues were isolated from flies infected with *T. brucei* strain 1738 dissected day 21 post infection (pi) and parasites from the salivary glands were additionally isolated at day 24 and 40 pi (**Fig 1A**). At days 21 and 24 pi, the tissues were incubated in media to release the free-swimming parasites, whereas at day 40 pi the cells were derived from free (spill-out from tissue) or attached (enzymatically disassociated) cell populations to capture metacyclics or attached epimastigotes and premetacyclics, respectively (**Fig 1A**). On day 24 pi, we aimed to analyse salivary gland parasites from an experimental cross between strains 1738 and J10. We isolated cells from single infections of each strain as well as co-infected tissue that contained both strains as well as a small number of hybrid progeny. We additionally used two cell preservation methods in the collection of these data to allow for more flexibility in processing time. Although small differences were observed in the number of genes detected between

preservation methods and live cells, this was confounded by timepoint, preventing us from fully understanding the impact of the preservation and handling techniques alone (**S3 Fig**).

To understand transcriptional variation at the single-cell level across tissues, strains, and time, we performed dimensionality reduction with all 388 cells using UMAP (**Fig 1B**). We observed that cells grouped by their tissue of origin, with clusters representing midgut and proventriculus trypanosomes and two groups of salivary gland parasites that we hypothesized could represent different cell-types or stages (**Fig 1B**). This idea was supported by the distribution of time points across the two salivary gland groups with 21- and 24-day pi cells distributed throughout the two groups, while the 40-day pi cells (squares) occupied the centre and right-hand area; the left-hand area therefore represents early salivary gland developmental stages, such as sexual stages, which are frequent at 21–24 days pi, but are relatively scarce at 40 days pi compared to epimastigotes and metacyclics [9].

We next used consensus clustering to partition the cells into six clusters based on the top average silhouette score in SC3 [32] (**Fig 1C**). The midgut and proventriculus parasites each formed a single cluster (clusters C1 and C2 respectively), whereas the salivary gland cells divided into four clusters (C3-C6). We identified 238 marker genes across the six clusters allowing us to assign potential cell types (AUROC > 0.75, $p < 0.01$, **Fig 1D** and **S1 Table**). C1 and C5 had very few significant marker genes and C6 had none (hence not included in **Fig 1D**). This is likely due to different overall levels of transcriptional activity across the cell types as we observed fewer genes per cell for these clusters (**S1 Table**). For further characterization, we additionally identified the top 200 genes expressed in each cluster (**S2 Table**). Based on both the cluster markers and top genes of each cluster, we were able to assign putative cell-types. C1 and C2 showed expression patterns consistent with midgut procyclics and proventricular forms, respectively, based on known marker genes and bulk transcriptomic data [21,33] (**S2 Table**). C1 was characterized by a single marker gene, *FHc* (Tb927.3.4500), a fumarate hydratase, which catalyses conversion of fumarate to malate in the TCA cycle (**Fig 1**) [4,34]. *FHc* was also the most significant marker gene for the midgut forms when integrated with *T. brucei* single-cell data from [20], supporting the cell-type assignment across datasets (**S4 Fig and S3 Table**). C1 also expressed several genes encoding surface proteins at high levels (*EP1*: Tb927.10.10260, *EP3-2*: Tb927.6.520, and *EP2*: Tb927.10.10250), as well as three Proteins Associated with Differentiation (*PAD1*: Tb927.7.5930, *PAD2*: Tb927.7.5940, *PAD7*: Tb927.7.5990) (**S2 Table**), which are implicated as sensors of environmental stimuli and trigger differentiation [35]. C2 had several marker genes associated with transport (*e.g.* amino acid transporter *AATP11* and purine nucleotide transporter *NT10*, **Fig 1**), both also highly expressed in C1 cells (**S2 Table**). Procyclin *EP2* was identified as a marker gene for this cluster, though both *EP1* and *EP3* (Tb927.6.480) were also highly expressed (**S2 Table**).

Cluster C3 comprised day 21/24 pi early salivary gland developmental stages (**Fig 1B and 1C**) including potential sexual forms. Notably, we observed high expression of the gamete fusion protein HAP2 (Tb927.10.10770), which is known to be expressed in meiotic intermediates and gametes (**S1 Table**) [10]. An analogous cluster was also identified by [20], which showed high expression of *HAP2* and *HOP1* (Tb927.10.5490), a meiosis-specific protein. Integration across these two datasets showed the cells from both studies clustered together at a granular level (**S4 Fig** and **S3 Table**). Other notable marker genes for this cluster were two leucine-rich repeat (LRR) protein genes (Tb927.9.14570, Tb927.7.7180), with a third also highly expressed (**S1 Table** and **S2 Table**). Procyclin genes *EP1*, *EP3* and *GPEET* (Tb927.6.510) (**S2 Table**) were highly expressed, together with *BARP* genes, which is the characteristic surface protein of epimastigotes [7].

Several *BARP* genes were the prominent marker transcripts in cluster C4 (**Fig 1D**), identifying this cluster as salivary gland epimastigotes, consistent with previous studies [19,20]. The

majority of the later time point (day 40 pi) cells were found in clusters C5 and C6 (**Fig 1B and 1C**), showing that these clusters represent the later salivary gland developmental stages including mature metacyclics and/or their immediate precursors, pre- and nascent metacyclics. However, the identities of these clusters remain unclear. C5 had only three significant marker genes, two of which encode zinc finger proteins and one hypothetical protein, and C6 none (**Fig 1D** and **S2 Table**). The zinc finger protein genes, *ZC3H11* (Tb927.5.810) and *ZC3H45* (Tb927.11.8470) were also identified as biomarkers of pre-metacyclics (Meta 1) in [19], and ZC3H11 was also identified as a highly expressed gene in purified, culture-derived mature metacyclics [27]. Several other highly expressed genes of mature metacyclics ([27]) were also identified in our data for either C5 or C6 (e.g. *ZFP2, HSP 110*; **S2 Table**). It is noteworthy that cluster C5 is predominantly strain J10, while cluster C6 is predominantly strain 1738 (**Fig 1D**), and hence these clusters may also represent strain-specific rather than stage-specific expression differences. Additionally, mature and nascent metacyclics both have VSG on the surface [36], and it is reasonable to suppose that pre-metacyclics already transcribe *VSG*s. In support of this, both previous scRNAseq studies found high levels of expression of *VSG*s in cells identified as pre-metacyclics (Meta 1, [19]; Pre-metacyclic, [20]). Here, *VSG*s were not identified as abundant transcripts in either C5 or C6, likely because of poor match between strain 1738/J10 *VSG* transcripts and the Tb927 reference genome, as *VSG* repertoires are strain-specific. We next endeavoured to identify the metacyclic *VSG* repertoire of strains 1738 and J10, in order to confirm the cell-types of C5 and C6.

## Expression of mVSG transcripts

Mature metacyclics can be unequivocally identified by their lack of expression of the cell surface proteins GPEET, EP and BARP, replaced by expression of a single *mVSG* gene in each cell [27], but pre-metacyclics also transcribe *mVSG*s at high level in [20]. To explore *mVSG* expression in salivary gland-derived cells, we first needed to identify the *mVSG* repertoires of strains 1738 and J10, which differ from those of previously published strains. We built a *de novo* transcriptome assembly based on all reads from the 388 tsetse-derived cells and identified putative *mVSG*s by comparison of the resulting ORFs to the total *VSG* repertoire of each strain, previously identified using a Hidden Markov Model on full-genome Illumina sequence data [37]. These assembled *mVSG* transcripts were then mapped to strain 1738 or J10 contigs to place them in a genomic context. Using this method, we identified 11 *mVSG*s that were expressed in our dataset all of which contained an upstream *mVSG* promoter based on the consensus sequence [38,39]. Additionally, downstream telomeric repeats were present in six of these contigs (**Fig 2A**). Although several *ESAG*s were found on the contigs, these were up- not downstream of the promotor and therefore not part of the *mVSG* expression site. The presence of these characteristic features of *mVSG* gave us further confidence that the identified transcripts were originating from *bona fide mVSG*. Interestingly, strains J10 and 1738 shared one *mVSG* (DN18105), suggesting some level of conservation across strains. This is the first identification of the *mVSG* repertoire in these strains.

Individual parasite transcriptomes were then mapped to the assembly to generate counts for each putative *mVSG*. *MVSG* transcripts were expressed by most cells in cluster C5, followed in order by C4, C6 and C3, with negligible expression in C1 and C2 (**Fig 2B and 2C**). Overall levels of expression were highest in cluster C5 (**Fig 2B–2D**), with 38% (16/42 cells) expressing more than one *mVSG* (**Fig 2B–2D**). Multiple *mVSG*s were also expressed by 38% (6/16) and 21% (4/19) cells in clusters C4 and C6 respectively, and a single cell in C3 (**Fig 2C and 2D**). Recent work by [20] confirmed expression of two *mVSG*s in pre-metacyclics using single molecule mRNA-FISH and put forward a model where multiple *mVSG*s are transcribed at low

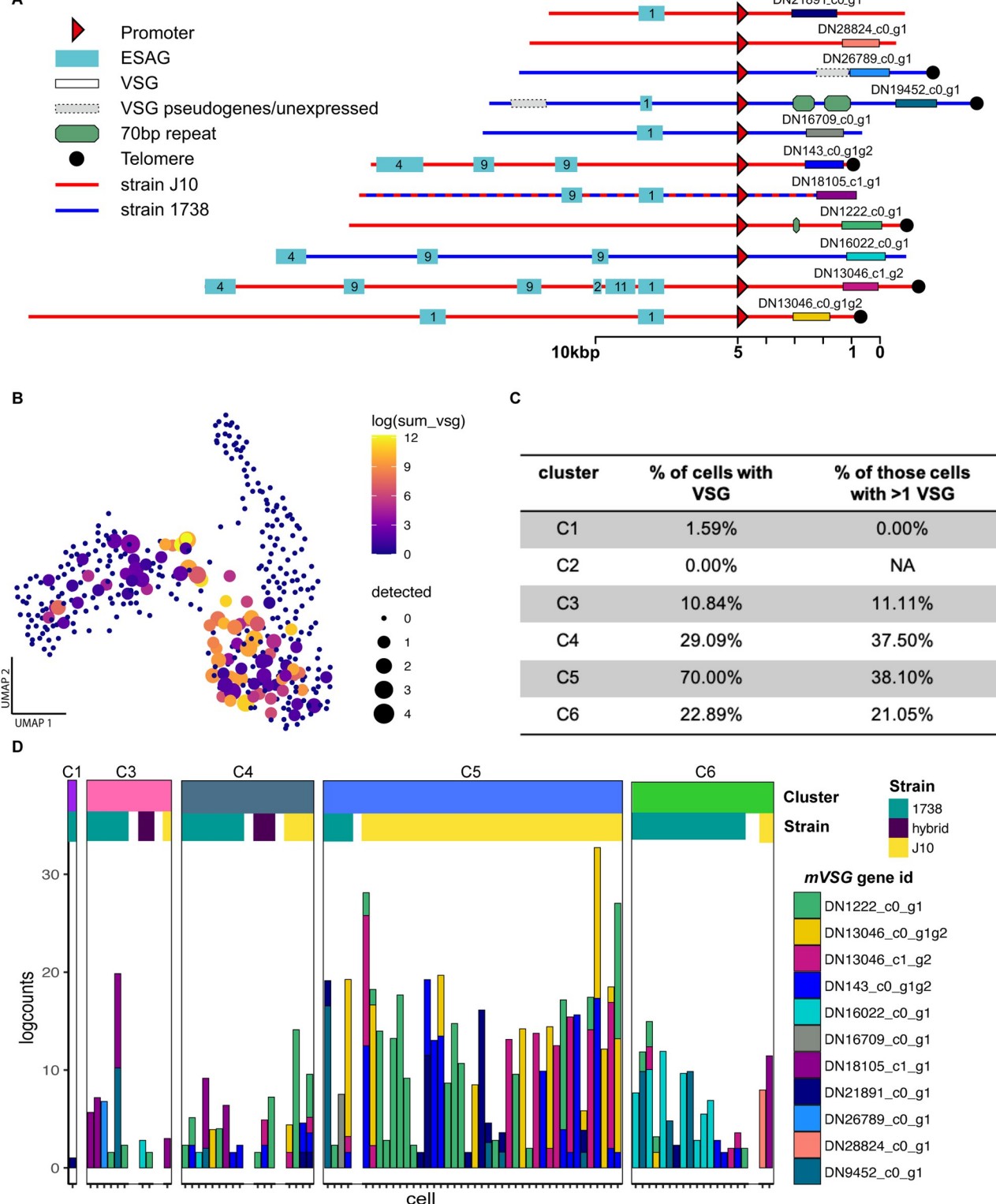

**Fig 2. mVSG expression in fly-derived trypanosomes.** (A) The genomic context of 11 *mVSG*s identified in strains 1738 and J10. The rectangles with a solid black outline represent the *mVSG* and are coloured to match Fig 2D. The sequences of each *mVSG* can be found in **S1 File**. (B) The transcript abundance of *mVSG* across 388 fly-derived trypanosome cells on the UMAP coloured by the logged sum of the *mVSG* counts in each cell and sized by the number of different *mVSG* detected in that cell. (C) The breakdown of *mVSG* expression per cluster. C5 had the highest proportion of cells expressing *mVSG* and the greatest proportion of those cells expressing multiple *mVSG*. (D) A barchart of all cells expressing *mVSG* (>1 read) organised by cluster and strain. Strain-specific expression of *mVSG* was seen at high levels in C5, which is primarily composed of strain J10.

levels initially, with a single *mVSG* dominating expression in the mature metacyclic forms. Based on this model, C4, C5 and C6 all contain a high proportion of pre-metacyclics, as well as some mature metacyclics. The *mVSG*s expressed varied over development and between strains, with DN1222 being the dominant transcript in C5, which were primarily strain J10 cells, and DN16022 being the dominant transcript in C6, which were primarily strain 1738 cells (**Fig 2D**). Observed expressed *mVSG*s are largely consistent with the *VSG*s present in their strain (when there is sufficient read depth), but low read count and partial coverage result in ambiguous assignment due to sequence conservation between *VSG*s.

## Pseudotime analysis of salivary gland development

To understand fine-scale changes in expression patterns during development of salivary gland parasites, we focused on the 161 salivary gland cells of strain 1738 collected over three time points (day 21, 24, 40 pi). The UMAP projection of these cells showed a general correspondence with the clusters identified in **Fig 1C**, with the left-hand group of cells representing clusters C3 and C4 (day 21/24 pi, early and late epimastigotes) and the right-hand group predominantly representing cluster C6 which included most of the day 40 pi transcriptomes (**Fig 3A and 3B**). The small branch connecting these two groups contained many cells collected from the dissociated salivary gland tissue, which likely represent attached epimastigotes and premetacyclics. Although we cannot rule out that these parasites were trapped unattached cells, their enrichment at this bottleneck in the UMAP indicates their importance in the developmental transition to metacyclics (**Fig 3C**).

We next used Slingshot [40] to temporally order these cells in pseudotime, revealing a trajectory running from left to right from gametes and early epimastigotes to metacyclics (**Fig 3D**). We discovered 691 genes that were differentially expressed over this trajectory (**Fig 3E and S4 Table**) and used hierarchical clustering to identify modules of co-expressed genes. Modules expressed early in development were enriched for genes involved in negative regulation of mitotic cell cycle and ATP metabolism (modules 2 and 13, **Fig 3E and S4 Table**), while middle-late modules were enriched for genes involved in translation and the ribosome, perhaps necessitated by the changes in surface proteins and metabolism associated with differentiation from epimastigotes to metacyclics (**Fig 3E and S4 Table**).

## Identification of hybrid progeny and strain-specific expression

To investigate potential cell-types and cell-type specific responses that could be involved in sexual reproduction at day 24 post-infection, we collected strains J10 and 1738 parasites from both single-strain infected and co-infected tsetse. In the co-infected treatment, we sorted cells from both strains based on fluorescence (strain 1738 GFP+, strain J10 RFP+), and sorted a small number of RFP+/GFP+ potential hybrid parasites (16 sorted, 14 passed QC). To confirm strain assignment, we used Souporcell to cluster different genotypes based on SNPs in the RNA-seq reads [41]. The two genotype clusters identified were each composed of one of the strains based on fluorescent identification with FACS (strain 1738 GFP+ = cluster 0; strain J10 RFP+ = 1), and the potential hybrid progeny were identified as inter-genotypic doublets (clusters 0/1 and 1/0), with alleles present from both strains (**Fig 4A**). This confirmed that six of the RFP+/GFP+ cells were genuine hybrids, while a further three hybrid cells were identified from the RFP-/GFP+ or RFP+/GFP- groups; as the GFP and RFP genes are present on only one homologue, four hybrid genotypes with respect to fluorescent protein genes are expected [42]. Looking at a UMAP of all day 24 pi cells, we observed some separation of strain in both the early and late epimastigote clusters (C3 and C4) and clusters C5 and C6 (**Fig 4B**). However, we did not see clear clustering based on the infection treatment (co- vs single-infection),

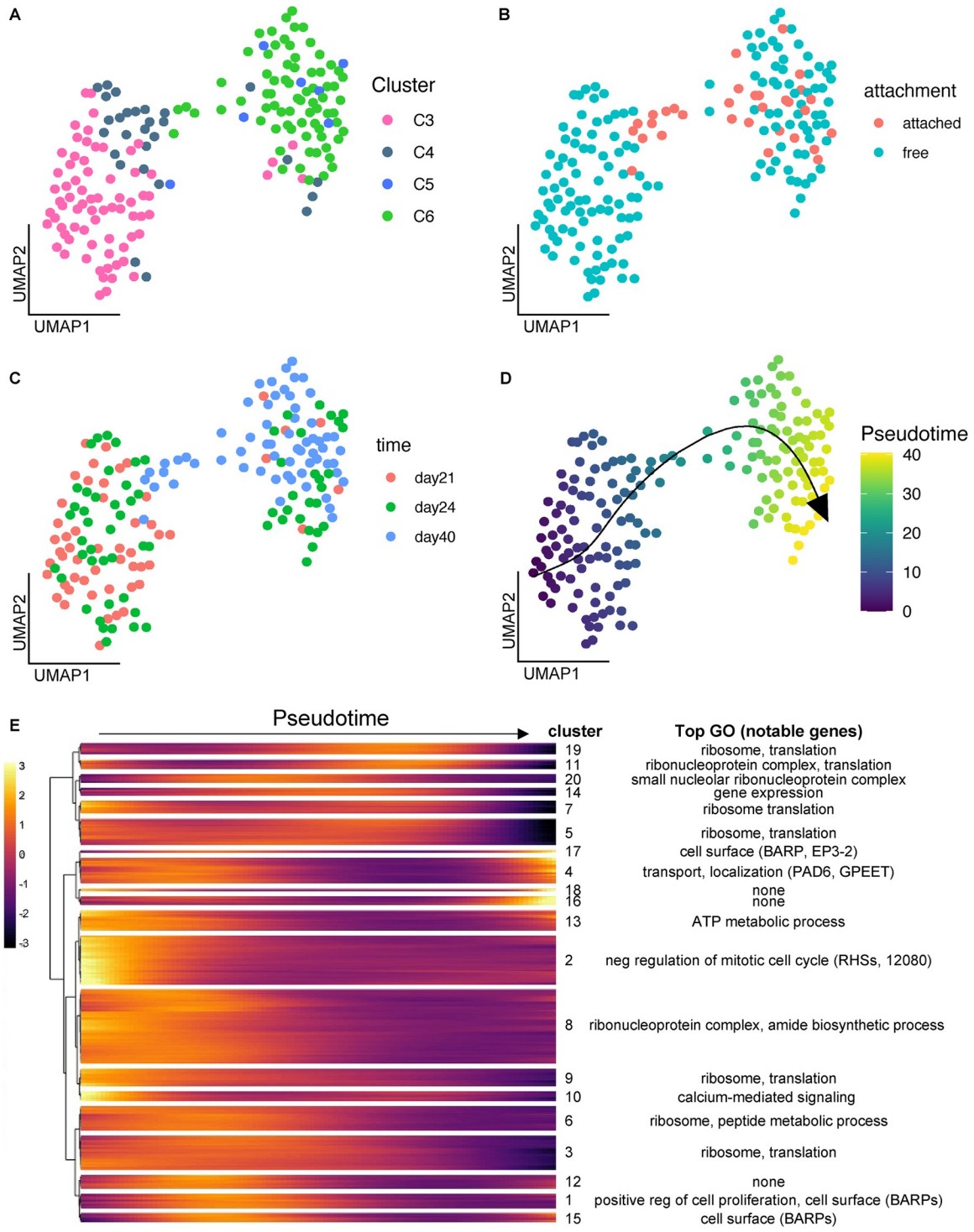

**Fig 3. Pseudotime trajectory analysis of developing salivary gland parasites.** Strain 1738 parasites collected from the salivary gland at day 21, 24 and 40 pi were used to map fine-scale changes in gene expression over development. (A-D) A UMAP of the 161 strain 1738 salivary glands parasites coloured by global cluster assignment from **Fig 1** (A), day PI (B), attachment treatment (C) and pseudotime assignment (D). (E) A heatmap of 20 clusters of genes differentially expressed over the pseudotime trajectory from (D).

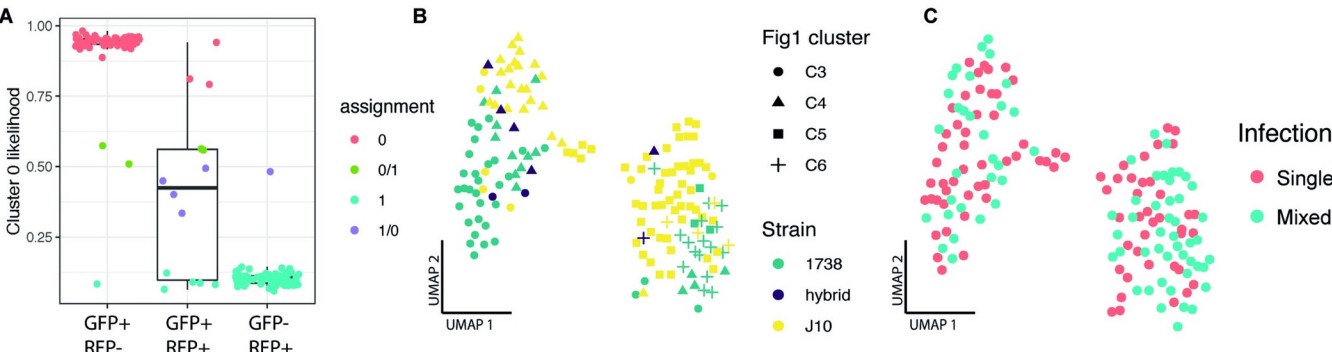

**Fig 4. Classification of hybrid progeny.** Souporcell was used to assign genotypes based on SNPs found between the two strains. The two genotype assignments (0,1) were each primarily composed of one of the strains based on fluorescent identification with FACS (strain 1738 GFP+/RFP- = cluster 0; strain J10 GFP-/RFP+ = 1), and the potential hybrid progeny were classified as inter-genotypic doublets (clusters 0/1 and 1/0). The likelihood ratio of cluster 0 assignment is shown for each of the three sorted populations (A). The UMAP of day 24 mixed- and single-infection experiments coloured by strain assignment and shaped by **Fig 1** cell cluster assignment (B) and infection treatment (C).

suggesting that there is no strong transcriptomic response to presence of another parasite strain (**Fig 4C**). In order to understand if the observed strain-specific clustering was a result of different cell-type composition or differential expression between strains within a cell-type, we integrated the data across strains using Seurat v3 [43]. Using this method, we were able to co-cluster the early epimastigote cells and identify 11 genes differentially expressed between the two strains (**S5 Fig and S5 Table**). However, the later stage cells seen in strain J10 had no representation of strain 1738, suggesting this cell-type is unique to this strain at day 24 pi, which could be observed if the strains have different developmental rates (**S5 Fig**).

## Transcript levels of procyclin and candidate novel sexual stage genes correlate with protein expression *in vivo*

A primary aim of this study was to identify the sexual stages of *T. brucei* and our results support the hypothesis that cluster C3 (**Fig 1**) represents meiotic intermediates and gametes, which are abundant around day 21 pi [8–10]. Looking at expression of genes encoding proteins known to be essential for sexual reproduction, we found high levels of expression of *HAP2* and also *GEX1* in cluster C3, with some signal from the meiosis-specific genes *DMC1* and *HOP1* (**Fig 5A**). Surprisingly, these cells also expressed the procyclin gene *GPEET*, which is considered to be a marker of early procyclics in the tsetse midgut, replaced by EP procyclins in late procyclics [31,44]. *GPEET*, *EP1*, *HAP2* and *GEX1* all have the highest expression in cluster C3 (**Fig 5B**). We used immunofluorescence to tie these observations to specific morphological forms and to validate the presence of GPEET on the surface of salivary gland parasites (**Fig 5C and S6 Table**). We found that GPEET, together with EP and BARP were present in >90% of the meiotic dividers (2K1N cell with large posterior nucleus and two flagella) and gametes (1K1N or 2K1N cells with small pear-shaped body and relatively long anterior flagellum) [9], and as expected absent in metacyclics (**Fig 5C**). Epimastigotes showed a similar pattern to the sexual forms but lower total proportions (**S6 Table**). Additionally, we looked at proventricular parasites (mesocyclics) and found expression of EP and GPEET but no BARPs, further confirming our gene expression data is matched at the protein level for these markers (**Fig 5C**).

Although previous work has shown that *GPEET* and *EP* transcripts are present in salivary gland parasites, expression of the corresponding proteins have not been demonstrated [45]. Instead, BARPs have been considered the major surface proteins in non-metacyclic, salivary gland forms [7]. Our data support a model where epimastigotes and sexual forms are

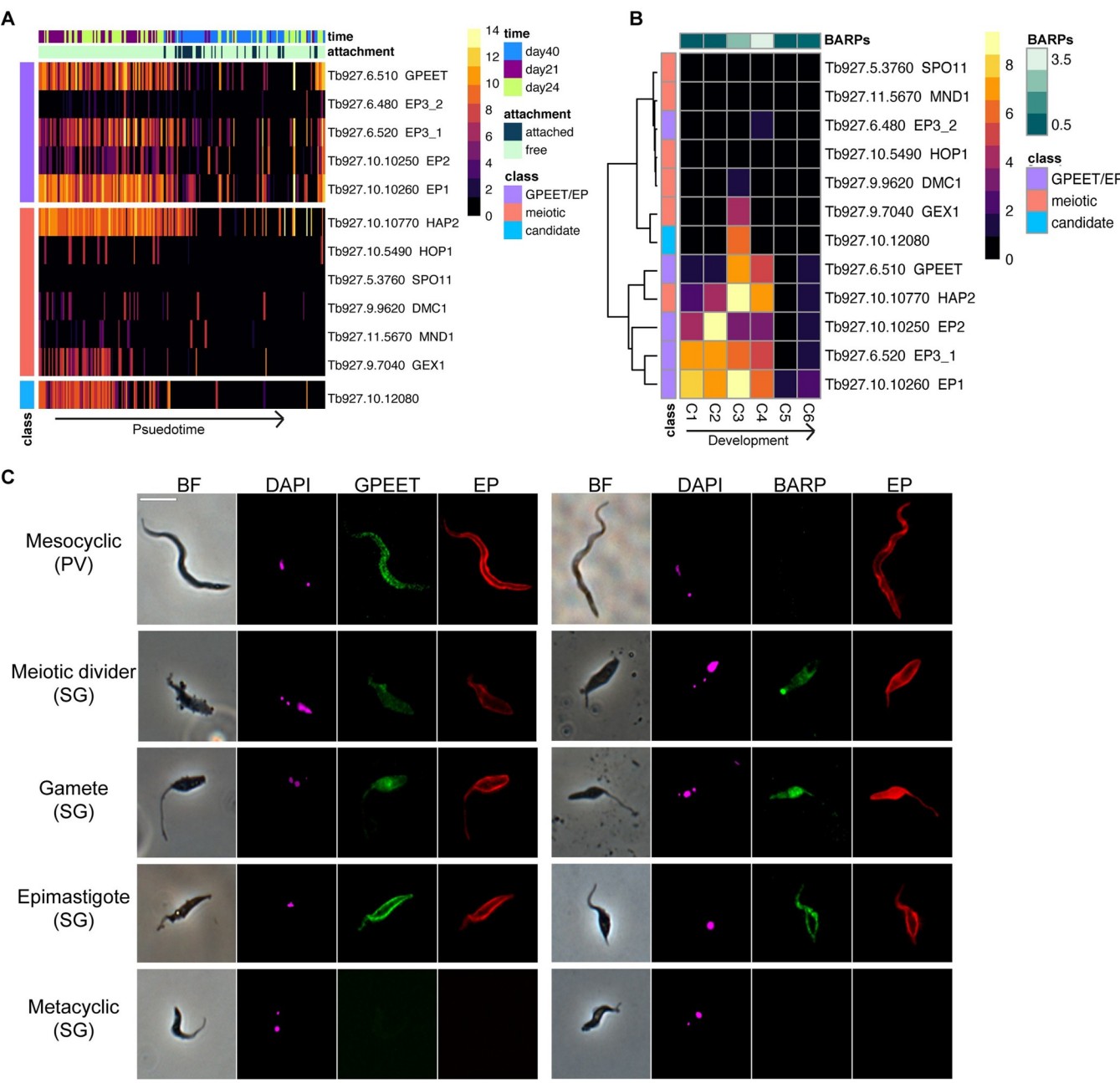

**Fig 5. Expression of surface antigens and genes involved in sexual reproduction throughout development in the tsetse fly.** We observed co-expression of procyclic surface antigen genes and *HAP2* in early parasite development in the salivary glands (A) and this general pattern of expression was also seen in proventricular forms (mesocyclics) (C2) as well as putative epimastigotes (C4) that also had high expression of *BARP*s (B). Immunofluorescence assays confirmed that these surface proteins corresponded to their transcriptional profiles and were present on the epimastigote and sexual stages (C).

expressing a diversity of surface proteins at high levels including BARPs, GPEET and EP. Additionally, the expression of these procyclins outside the midgut indicates they might play a functional role in these stages as well, and further experiments are needed to understand if they are required for development or interactions of meiotic dividers and gametes.

Cluster C3 was additionally characterised by strong and unique expression of Tb927.10.12080 (**Figs 1D, 5A, 5B, and 6A**), and we hypothesized that this gene may play a

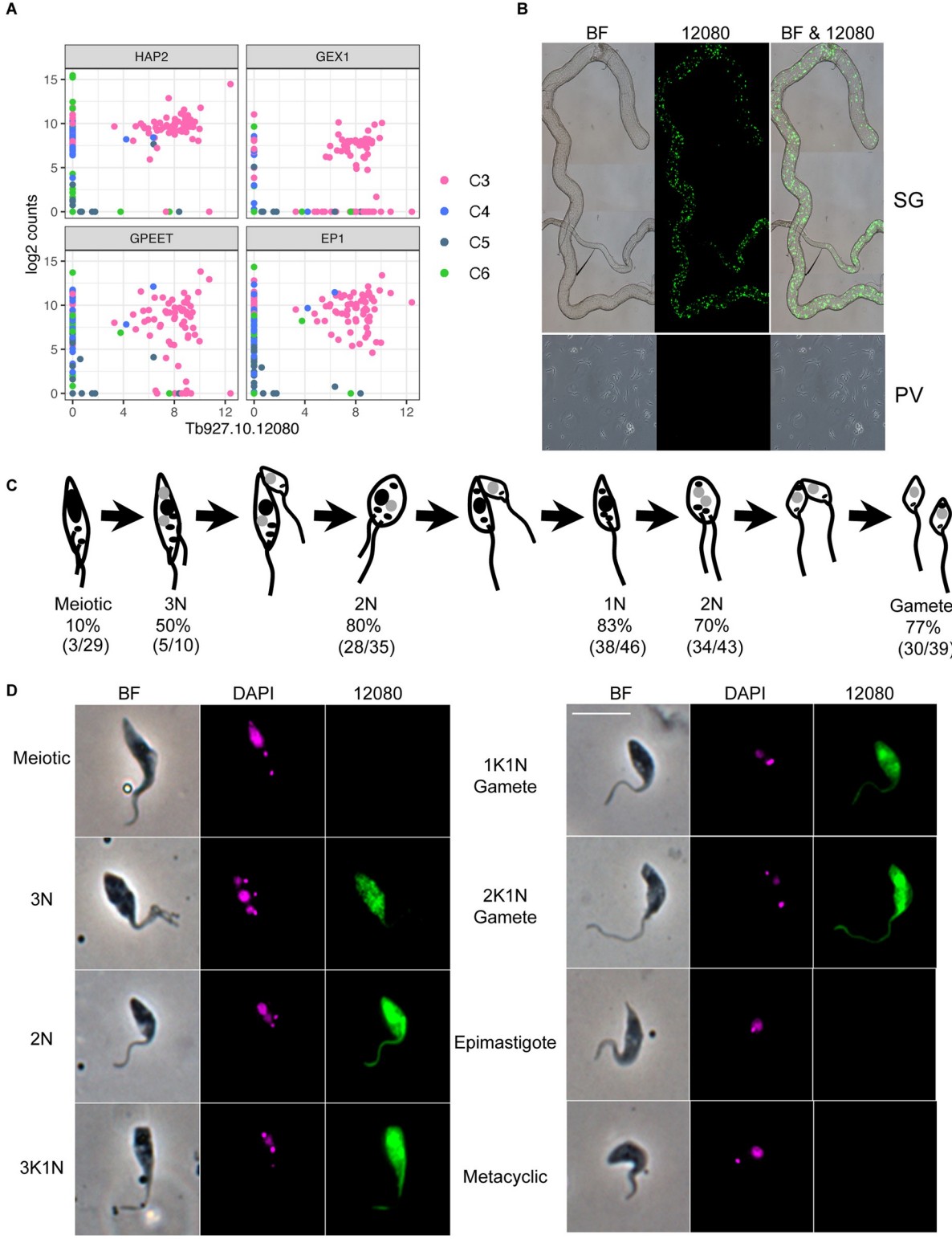

**Fig 6. Expression of Tb927.10.12080 coincides with sexual forms.** (A) Co-expression of Tb927.10.12080 with genes encoding HAP2 and GEX1, proteins associated with gamete and nuclear fusion in eukaryotes, and the surface antigen genes GPEET and EP1; co-expression is seen in a subset of cells from C3 (Fig 1D). (B) *T. brucei* strain 1738 GFP::Tb927.10.12080–3'UTR transcribed from the procyclin promotor in the salivary gland (SG) and proventricular forms (PV). (C) Diagram showing major cell types observed during meiosis in *T. brucei* (adapted from [10]); nuclei are shown in black (4C or 2C DNA contents) or grey (1C, haploid) and kinetoplasts are shown as smaller black dots.

Values beneath are the numbers and percentages of cells recorded for each cell type; both 1K1N and 2K1N gametes are included in the gamete total. Full data are presented in **S7 Table**. (D) Trypanosomes from tsetse fly salivary gland spill-out 16–21 days pi with *T. brucei* strain 1738 expressing GFP::Tb927.10.12080–3'UTR transcribed from the procyclin promotor. Left to right: phase contrast, DAPI, GFP:: Tb927.10.12080–3'UTR. The scale bar represents 10 μm.

role in sexual development. It encodes a hypothetical protein devoid of recognisable functional domains that is well-conserved in other trypanosomes including *T. congolense*, *T. vivax*, *T. cruzi* and *T. grayi*, and the C-terminal domain in more distantly related members of the trypanoso- matid family such as *Leishmania* spp. The gene falls upstream of two RNA binding proteins (Tb927.10.12090 (*RBP7a*); Tb927.10.12100 (*RBP7b*)) and the recently identified long non-cod- ing RNA, *grumpy*, which all play a role in stumpy form differentiation [46,47]. Along with Tb927.10.12080, this genomic region could potentially act as a hotspot for differentiation and developmental processes across the parasite life cycle. Localisation data for Tb927.10.12080 from Tryptag.org [48,49] is equivocal, showing punctate cytoplasmic fluorescence for the N-ter- minal tagged protein and a mitochondrial location in a proportion of cells for the C-terminal tagged protein. To investigate expression of this protein during development in the tsetse fly, we used the 3' UTR to regulate expression of GFP driven by the procyclin promotor (**S6 Fig**). At 20–22 days pi, there was very little detectable expression in midgut procyclics or proventricular forms, but strong expression in salivary gland trypanosomes (**Fig 6B**). Overall, of the parasites from the salivary gland scored, 39% (182/464) showed expression (**S7 Table**). These included meiotic intermediates and both 1K1N and 2K1N gametes but not metacyclics or unattached epimastigotes (**Fig 6C and 6D**). Cell types involved in the early stages of meiosis, such as mei- otic dividers and 3N cells with one diploid and two haploid nuclei [10], had lower percentages of cells expressing (10% and 50% respectively) than did those involved in the later stages of mei- osis and in gametes (77–80%; **Fig 6C and S7 Table**). At 37–38 days pi, the percentage of fluores- cent trypanosomes dropped (103/682; 15%) and these cells were misshapen with no recognisable gametes or sexual intermediates. Further experiments are needed to understand the functional role of Tb927.10.12080 in meiosis and sexual development; however, it's unique pattern of transcript and protein expression indicate it could play a vital role in processes that allow for genetic exchange in *T. brucei* and perhaps more broadly in kinetoplastids.

## Conclusion

Here we applied single-cell RNA sequencing to explore the heterogeneous trypanosome popu- lations in the tsetse fly. These data provide a resource for the parasitology community, which we have made available via an interactive website (http://cellatlas.mvls.gla.ac.uk/). Addition- ally, this data set allowed us to elucidate the transcriptional profiles of key life cycle stages in the salivary glands including the sexual stages. From this mixture of cell types, we were able to identify a cluster of cells that shared a particular transcriptomic profile characterized by high expression of the gene encoding the gamete fusion protein HAP2, together with several unstudied genes. One of these was a kinetoplastid-conserved gene Tb927.10.12080, which was exclusively expressed at high levels by meiotic intermediates and gametes. We speculate that this protein, currently of unknown function, plays a role in gamete formation and/or fusion, but further experiments are needed to test this hypothesis.

## Materials and methods

### Data collection

**Trypanosome culture and tsetse infection.** The following tsetse-transmissible strains of *Trypanosoma brucei brucei* were used: *T. b. brucei* strain J10 (MCRO/ZM/73/J10) and strain

1738 (MOVS/KE/70/EATRO 1738); each was genetically modified to express a fluorescent protein gene (strain J10 RFP, strain 1738 GFP). Mating between these strains has been demonstrated previously [42] and strain 1738 reliably produces large numbers of gametes around day 21 post-infection [9]. Procyclic form (PF) trypanosomes were grown in Cunningham's medium (CM) [50] supplemented with 15% v/v heat-inactivated foetal calf serum, 5 μg/ml hemin and 10 μg/ml gentamycin at 27˚C. Tsetse flies (*Glossina pallidipes*) were infected with PF trypanosomes, maintained and dissected as described previously [8].

**Parasite isolation from tsetse tissues for scRNAseq.**   Free swimming parasites were obtained from *G. pallidipes* by separately pooling tissues into CM (5 midguts in 500 μl CM; 5 proventriculi in 50 μl CM; 20 sets of salivary glands in 50 μl CM). Tissues were incubated at room temperature (RT) for 10 minutes prior to filtration through a 100 μm filter. Cells were washed once with 1 ml CM prior to preservation or sorting. At day 40 pi, the parasites attached to the salivary glands were isolated by disassociation of the tissue after the 10-minute incubation period. Forceps were used to transfer the tissue to an enzymatic solution consisting of 200 μl Collagenase IV (1 mg/ml) and 25 μl of Elastase (4 mg/ml). The sample was then incubated at 30˚C for 40 min with shaking at 300 rpm. During the incubation, the tissue was disrupted by pipetting up and down 40 times every 15 minutes at first with a p1000 pipette set to 150 μl and then with a p200 set to 100 μl once the tissue started to break up.

**Cell preservation.**   A subset of cells was preserved prior to cell sorting to allow for greater flexibility in the time between collections and FACS. The day 40 pi salivary glands parasites (attached and free) cells were fixed by adding 200 μl of dithio-bis(succinimidyl propionate) (DSP; Lomant's reagent) dropwise to the cell pellet as described in [51]. DSP fixed samples were incubated at room temperature for 30 minutes prior to adding 4 μl 1 M Tris-HCl pH 7.5. Samples were then stored at 4˚C for up to 24 hours. Prior to sorting, DTT was added to a final concentration of 50 mM. The day 24 pi salivary gland parasites (cross vs single infection) were preserved by resuspending the cell pellet in 200 μl Hypothermosol-FRS (BioLifeSolutions) [52]. Samples were then stored at 4˚C for 5 hours prior to sorting. Live cells were stored at 4˚C after collection from the tsetse tissue for up to 5 hours prior to sorting.

**Cell sorting, library preparation and sequencing.**   All parasite cells were sorted within 24 hours of collection on an Influx cell sorter (BD Biosciences) with a 200 μm nozzle or a Sony SH800 with 100 μm chip. Parasites were sorted based on RFP and/or GFP fluorescence into nuclease-free 96 well plates containing lysis buffer as described previously [12]. Sorted plates were spun at 1000 g for 10 seconds and immediately placed on dry ice. Reverse transcription, PCR, and library preparation were performed as described in [12]. Cells were multiplexed to 384 and sequenced on a single lane of Illumina HiSeq2500 v4 with 75 bp paired-end reads.

**Immunofluorescence.**   Salivary glands, proventriculi and midguts from infected flies (20–22 days pi) were pooled separately into CM and incubated at RT for 10 minutes (to allow trypanosomes to swim out of tissue) and then filtered through a 100 μm filter with PBS. Trypanosomes were pelleted by centrifugation and resuspended in 100 μl PBS. Cells were fixed overnight at 4˚C by adding 100 μl 6% paraformaldehyde, 0.1% glutaraldehyde in PBS, and then washed twice with PBS before resuspension in 50 μl PBS. Cell suspensions were pipetted onto 2 x 10 mm coverslips, allowed to settle for 20 mins in a humid chamber, and then liquid was removed and replaced by 2% BSA in PBS. After 30 mins liquid was removed and cells incubated with 2% BSA in PBS containing diluted antibody for 30 mins at RT. Rabbit anti-GPEET (1:1000) and rabbit anti-BARP (1:1000) were a kind gift from Isobel Roditi, University of Bern, Switzerland; mouse anti-EP mAB (1:100) was from Cedarlane. Cells were washed three times with PBS and incubated with 2% BSA in PBS containing anti-rabbit FITC (1:1000) and anti-mouse TRITC (1:1000) for 30 mins at RT. Cells were washed three times with PBS, briefly air dried, stained with DAPI in VECTASHIELD mounting medium (Vector

Laboratories) and viewed using a DMRB microscope (Leica) equipped with a Retiga Exi camera (QImaging) and Volocity software (PerkinElmer). The whole area of the coverslip was scanned systematically from top to bottom, capturing FITC, TRITC, DAPI and phase contrast images of each trypanosome. Digital images were analysed using ImageJ (http://rsb.info.nih.gov/ij).

**Tb927.10.12080 gene expression.** The 3' UTR of Tb927.10.12080 was amplified from genomic DNA of strain 1738 using the primers 5'- GATCCTCGAGTAGTGGCGAGTGTT TACAACAGTGTC and 5'-GATCGGGCCCCTTGTGCGGATCCAAACAA, and inserted immediately downstream of a GFP gene driven by the procyclin promotor from plasmid backbone pHD449, which is designed for insertion into the tubulin locus (**S6 Fig**) [53,54]. The plasmid construct was used for stable transfection of procyclic strain 1738 and following hygromycin selection, clones were tsetse fly transmitted as described by [8,9]. Flies were dissected 10–40 days pi and organs viewed by fluorescence microscopy and imaged live or fixed in 2% paraformaldehyde and stained with DAPI in VECTASHIELD mounting medium (Vector Laboratories).

## scRNAseq data analysis

**Mapping and generation of expression matrices.** Nextera adaptor sequences were trimmed from fastq files using trim_galore (*-q 20 -a CTGTCTCTTATACACATCT—paired—stringency 3—length 50 -e 0.1)* (v 0.4.3) [55]. Trimmed reads were mapped using HISAT2 (*hisat2—max-intronlen 5000 -p 12)* (v 2.1.0) [56] to the *T. b. brucei* 927 genome. The GFF was converted to GTF using the UCSC genome browser tool [57]. Reads were then summed against genes using HTseq (*htseq-count -f bam -r pos -s no -t CDS)* (v 0.7.1) [58].

**Assembly of VSG transcripts.** Because there is a lack of conservation of *VSG*s across *T. brucei* strains, we built a *de novo* transcriptome assembly to identify the *mVSG* transcripts expressed in strains 1738 and J10. First, we merged the BAM files across the 388 cells and converted to FASTQ using bedtools (v. 2.29.2) [59]. Using Trinity (v. 2.1.1) [60] to assemble the transcripts from this merged file, we detected 53521 'genes' with a mean contig length of 800 bp. We then mapped each cell to this assembly using RSEM (v. 1.3.3) to generate a counts matrix and used Transdecoder (v. 5.5.0) to detect open reading frames [61]. BLASTp (v. 2.9.0) was used to match to putative *VSG*s that had been curated independently from whole genome data as described below. Transcripts with >90% identity were used for further analysis.

Genomes for the parent strains J10 and 1738 were assembled from 76 bp paired read Illumina data from [62] using SPADES under default parameters [63]. Predicted *mVSG*s were identified to genomic loci, using BLAST against the assembled contigs with a percent identity across the entire transcript >95%, alignments of a raw score of greater than 1000 were further investigated. Additional open reading frames were identified by BLAST alignment of the curated 927 annotated CDS set. Nhmmer was used to identify putative *mVSG* promoters from the alignments [38,39].

**Quality control and normalization.** Quality control and visualisation was performed in Scater (v. 1.12.2) [64]. Cell quality was assessed based on the distribution of genes detected per single-cell transcriptome. Cells with fewer than 40 genes or more than 3000 genes detected were removed, as well as cells that had fewer than 1000 total reads. These QC thresholds allowed us to keep more cells in the analysis that are likely to be less transcriptionally active such as mature metacyclics. Out of 515 parasites isolated from tsetse tissue that were sequenced, 388 passed quality control and were used for downstream analyses. Raw count data was normalized using a deconvolution size factor in Scran (v. 1.16.0) [65] to account for differences in overall level of expression between cell-types.

**Cell clustering, projection, and marker genes.** In order to unbiasedly group transcriptomes based on similar expression profiles, 388 cells collected from the tsetse were clustered using K-means clustering in SC3 (v. 1.12.0) [32]. Dimensionality reduction was performed in Scater (v. 1.12.2) [64] using UMAP with the top 200 most variable genes and n_neighbors = 5, min_dist = 1, spread = 3. Marker genes were identified for each cluster using SC3 (AUROC >0.75 & adjusted $p$-value < 0.01).

**Identification of hybrid parasites and data integration.** To select different parasite genotypes (strains J10, 1738, or J10x1738 cross) in the mixed infection treatment, we first FACS sorted based on GFP+ (strain 1738), RFP+ (strain J10), or GFP+/RFP+ (hybrid) expression. We then used souporcell (*-k 2 -p 2*) (v2.0) [41] to confirm genotype assignment based on SNP profiles from the scRNAseq reads. Souporcell uses mixture model clustering to identify genotypes and potential cell doublets after SNPs are called and counted with freebayes and vartrix, respectively [41,66,67]. We identified hybrid parasites as those that were categorised as doublets (containing alleles from both strains).

To identify genes that were differentially expressed between the two strains, we used Seurat (v3.1.5) to integrate the data by identifying anchors with the FindIntegrationAnchors() and IntegrateData() functions. The data was then clustered using FindNeighbors() with the top 30 principal components and FindClusters() with a resolution of 0.5. Differential expression was performed using the FindMarkers() function (adjusted $p$-value < 0.001). The same integration methods were used to compare the data to [20] except that the top 20 principal components were used, the cluster resolution was 0.8 and the FindConservedMarkers() function identified markers found in both studies (max $p$-value < 0.001).

**Pseudotime and differential expression.** To assess developmental progression in strain 1738 salivary gland parasites from day 21-, 24-, and 40-days pi, Slingshot (v. 1.8.0) was used to estimate pseudotime by first inferring the global lineage structure of the cells and then fitting a smooth curve to infer the pseudotime variables for cells along that lineage [40]. Input clusters (k = 4) and the UMAP reduction were initially identified in SC3 and scater, respectively [32,64]. Genes differentially expressed over this trajectory were identified using the associationTest() function in TradeSeq (v. 1.4.0) [68].

## Supporting information

**S1 Fig. Quality assessment and expression of marker genes in procyclic culture singe-cell transcriptomes.** Forty-eight transcriptomes were generated using Smart-seq2 from parasites in a procyclic culture including a no cell and ten cell control. **(A)** The distribution of the total counts and total features (genes) detected in these 48 transcriptomes. **(B)** The total features plotted against total counts for the 46 single-cell transcriptomes shows a plateau as features and counts increase, suggesting that sequencing was saturated for these cells. We detected a mean of $2.6 \times 10^6$ reads and 1756 features per single-cell transcriptome. **(C)** Expression of procyclic surface antigen genes *GPEET* (Tb927.6.510), *EP1* (Tb927.10.10260), *EP2* (Tb927.10.10250), *EP3_1* (Tb927.6.520), *EP3_2* (Tb927.6.480). (TIF)

**S2 Fig. Quality control of insect stage parasites.** The distribution of genes detected (left) and counts (right) in each cell across the three insect tissues: midgut (MG), proventriculus (PV), and salivary glands (SG). Cells with fewer than 40 or more than 3000 genes per cell were removed. Additionally, cells with fewer than 1000 reads were removed. Cut-offs are represented by the red vertical lines in each histogram. After QC we detected a mean of 889 genes per cell and $1.1 \times 10^5$ counts per cell. (TIF)

**S3 Fig. Assessment of preservation methods of salivary gland parasites.** The distribution of features detected in salivary gland cells across the two preservation treatments (DSP and hypothermosol) compared to live parasites. Although there were slight differences in detection between the different treatments, caution must be taken in interpreting these differences as the fixation methods are confounded with the different timepoints collected (DSP: day 40; hypothermosol: day 24; live: day 21).
(TIF)

**S4 Fig. Integration with Hutchinson dataset.** All 388 tsetse transcriptomes were integrated with the Hutchinson dataset collected from salivary glands [20] using Seurat's data integration function. Plots show the UMAP of integrated data coloured by study **(A),** cluster identity from the different studies (paper_id) **(B),** integrated cluster assignment **(C),** or gene of interest **(D-F).** *FHc* (Tb927.3.4500) was the top marker gene (based on adjusted p-value) for the midgut and proventricular form cluster 4 **(D).** Tb927.7.380 (hypothetical protein, conserved) was the top marker gene for cluster 3 which contained gamete and epimastigote forms. *HAP2* (Tb927.10.10770) **(F)** was not a marker gene for the gamete cluster likely because of its ubiquitous expression across non-metacyclic forms. Although we were able to identify conserved marker genes across the two studies, separation remained in the UMAP for all cell-types **(A-B)** and only the non-metacyclic forms co-clustered across the two studies and only at a granular level. The metacyclic forms likely did not cluster together because of different VSG repertoires, and the separation across other cell-types may be due to time point, strain-specific expression patterns, or collection methods. Conserved marker genes for clusters 3 and 4 can be found in **S3 Table.**
(TIF)

**S5 Fig. Strain-specific gene expression (A).** A UMAP of the day 24 SG parasites integrated by strain (1738 and J10). Points are coloured by strain and shaped by **Fig 1** cluster. **(B).** The integrated UMAP coloured by new cluster from the integration analysis. Cluster 0 has a representation of both strains, whereas cluster 1 and 2 are composed primarily of strain 1738 or J10, respectively. **(C)** Differential expression was performed between strains within cluster 0. The ten genes differentially expressed between the two strains are displayed on a heatmap.
(TIF)

**S6 Fig. Plasmid map for GFP::Tb927.10.12080–3'UTR construct.** Life cycle selective expression of Tb927.10.12080 was investigated through a reporter construct where the expression of GFP was controlled by ~500 bp of UTR downstream of the gene. For this study a stable transformant line was generated in strain 1738 using the 3' UTR from its endogenous gene and integrated into the tubulin locus.
(TIF)

**S1 Table. Marker genes from Fig 1.**
(XLSX)

**S2 Table. Top 200 genes expressed in each cluster from Fig 1.**
(CSV)

**S3 Table. Marker genes from integration with Hutchinson dataset.**
(XLSX)

**S4 Table. Genes differentially expressed over pseudotime (corresponding to Fig 3).**
(XLSX)

**S5 Table. Genes differentially expressed between strains (corresponding to S5 Fig).** (CSV)

**S6 Table. Surface protein expression of trypanosomes from salivary glands from tsetse dissected 19–21 days post infected feed, using immunofluorescence.** (DOCX)

**S7 Table. Cell types recovered from tsetse salivary gland exudate 16–21 days post infection with *T. brucei* strain 1738 expressing GFP::Tb927.10.12080–3'UTR scored for GFP fluorescence.** (DOCX)

**S1 File. Fasta file of *mVSG* sequences.** (TXT)

## Acknowledgments

We gratefully acknowledge the generous supply of tsetse pupae from the International Atomic Energy Agency, Vienna. We thank the Sanger Institute flow cytometry core facility for technical support and guidance. We thank Thomas Otto and Jesse Ropp for development and deployment of the interactive website.

## Author Contributions

**Conceptualization:** Virginia M. Howick, Lori Peacock, Chris Kay, Wendy Gibson, Mara K. N. Lawniczak.

**Data curation:** Virginia M. Howick, Lori Peacock, Chris Kay, Wendy Gibson.

**Formal analysis:** Virginia M. Howick, Lori Peacock, Chris Kay, Wendy Gibson.

**Funding acquisition:** Virginia M. Howick, Wendy Gibson, Mara K. N. Lawniczak.

**Investigation:** Virginia M. Howick, Lori Peacock, Chris Kay, Clare Collett, Wendy Gibson.

**Methodology:** Virginia M. Howick, Lori Peacock, Chris Kay, Clare Collett, Wendy Gibson.

**Resources:** Wendy Gibson, Mara K. N. Lawniczak.

**Supervision:** Wendy Gibson, Mara K. N. Lawniczak.

**Validation:** Lori Peacock, Chris Kay, Wendy Gibson.

**Visualization:** Virginia M. Howick, Lori Peacock, Chris Kay, Wendy Gibson.

**Writing – original draft:** Virginia M. Howick, Lori Peacock, Chris Kay, Wendy Gibson.

**Writing – review & editing:** Virginia M. Howick, Lori Peacock, Chris Kay, Clare Collett, Wendy Gibson, Mara K. N. Lawniczak.

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
