## [Decision Letter · Decision Letter 0]

3 Dec 2021

Dear Dr Howick,

Thank you very much for submitting your manuscript "Single-cell transcriptomics reveals expression profiles of Trypanosoma brucei sexual stages" for consideration at PLOS Pathogens. As with all papers reviewed by the journal, your manuscript was reviewed by members of the editorial board and by several independent reviewers. The reviewers appreciated the attention to an important topic. Based on the reviews, we are likely to accept this manuscript for publication, providing that you modify the manuscript according to the review recommendations.

I am returning your manuscript with three reviews. After reading the manuscript and the reviews, all the comments can be addressed without further experimentation. However, several points highlighted by the reviewers must be addressed.

-Please pay particular attention to the language used to describe the role of Tb927.10.12080.

-Please consider the addition of data mentioned in the materials and methods to Figure 6 to further strengthen the interpretation of the role of Tb927.10.12080.

-All three reviewers have highlighted sections in the text that need to be expanded for clarity or typos addressed

Sincerely,

Lucy Glover

Guest Editor

PLOS Pathogens

David Horn

Section Editor

PLOS Pathogens

Kasturi Haldar

Editor-in-Chief

PLOS Pathogens

orcid.org/0000-0001-5065-158X

Michael Malim

Editor-in-Chief

PLOS Pathogens

orcid.org/0000-0002-7699-2064

I am returning your manuscript with three reviews. After reading the manuscript and the reviews, all the comments can be addressed without further experimentation. However, several points highlighted by the reviewers must be addressed.

Please pay particular attention to the language used to describe the role of Tb927.10.12080.

Please consider the addition of data mentioned in the materials and methods to Figure 6 to further strengthen the interpretation of the role of Tb927.10.12080.

All three reviewers have highlighted sections in the text that need to be expanded for clarity or typos addressed

These revisions must be addressed in order to prepare this manuscript for publication.

Reviewer Comments (if any, and for reference):

Reviewer's Responses to Questions

**Part I - Summary**

Reviewer #1: The manuscript by Howick et al., “Single-cell transcriptomics reveals expression profiles of Trypanosoma brucei sexual stages”, is an important contribution to the field of trypanosome development in the tsetse vector. The authors define and characterize different life-cycle stages transcriptomes, and, combined with previous studies using scRNAseq improve our understanding of the intricate development of T. brucei. The available repertoire of metacyclic VSGs (mVSGs) is expanded by 11 new genes which the authors identified in the cell lines used. The focus of the study is the sexual development of T. brucei, a process which is still not well understood mechanistically. A new marker for meiotic stages was identified and verified using a fluorescent reporter. The manuscript is well-written and is an important step to understanding meiosis and sexual reproduction in T. brucei, as well as its evolution in eukaryotes.

Reviewer #2: In this manuscript, Howick et al use single cell transcriptomics to study sexual recombination in African trypanosomes. The authors obtained single cell transcriptomes for trypanosomes from the midgut (MG), proventriculus (PV) and salivary gland (SG) of infected tsetse flies, covering the major stages of development in the insect host. The range of data, covering three organs, attached and free SG parasites, multiple strains, experimental crosses, assembly of metacyclic VSG expression sites, and the first description of the surface coat of meiotic and gamete cells is impressive.

These data therefore go further than the recent salivary gland single cell transcriptomes from Vigneron et al (2020) and Hutchinson et al (2021), which focused on the salivary glands only. The focus on sexual stages in this manuscript gives important novelty over these previously published transcriptomes.

This manuscript should be published without further experimentation. It will catch the attention not only of parasitologists, but also of people interested in the history of sexual recombination. We believe it could benefit from the addition of some data (from existing experiments but not shown), and of a more elaborate discussion. This one is a bit short but this is probably a consequence of the results and discussion being combined in a single section.

Reviewer #3: The manuscript describes single cell transcriptomes of trypanosomes from the salivary glands of tsetse flies. The findings are interesting and are confirmed by both the know biology in the salivary gland and by experimentation. A good read.

**Part II – Major Issues: Key Experiments Required for Acceptance**

Reviewer #1: Major points:

My main concern stems from the use of the following phrases to describe Tb927.10.12080:

p. 35 “We speculate that this protein… plays a role in gamete formation and/or fusion.”

p. 369 “…we hypothesized that this gene may play a role in sexual development.”

p. 394 “…it could play a vital role in processes that allow for genetic exchange in T. brucei…”

p. 421 “We speculate that this protein… plays a role in gamete formation and/or fusion.”

The results presented clearly demonstrate that the protein is expressed and likely functions during these developmental stages. However, speculating that this gene plays a direct role in the process is not justified. To demonstrate a functional role for Tb927.10.12080 in the sexual development of T. brucei the authors can use the same plasmid vector they used for the fluorescent reporter, but instead of the reporter, have a long hairpin construct for RNAi targeting the gene transcript. Constitutive expression should not be a problem because the protein does not seem to be expressed in other life-cycle stages in the fly. If RNAi against Tb927.10.12080 interferes with meiosis, scored by cell morphology/KN counts, the authors will have a very strong case for a direct role of this gene in T. brucei sexual development.

Alternatively, the language can be changed to indicate that there is a distinction between “functioning during” and “playing a role in”.

With the identification of additional 11 mVSG sequences to complement the set of already available metacyclic VSGs from different T. brucei cell lines, have the authors identified any features that distinguish these VSGs from the ones found in bloodstream expression sites? It is an important general question that still awaits clarification. The predominant current view is that they are indistinguishable from each other, however, it is important to comment on this point.

There should be a slightly expanded discussion/interpretation of the results indicating that GPEET procyclin is highly expressed in meiotic intermediates and gametes. GPEET is a presumed marker for early procyclics, as the authors point out. Or is this notion incorrect, according to the results presented here?

Reviewer #2: No new experiments required

Reviewer #3: In places the manuscript will be difficult to follow without knowledge of the software, the are a couple of instances of software being used without providing a few words on what type of analysis it does. It could be made more accessible for a wider audience.

Could the authors provide an estimate of the time from removing the cells from their natural habitats and lysing / fixing them for RNA preparation? As the mean half life of mRNAs is ~20 minutes in trypanosomes, a consideration of / comment on any possible effect of the cell handling on transcriptome would be helpful.

**Part III – Minor Issues: Editorial and Data Presentation Modifications**

Reviewer #1: Minor point:

1. In Fig. 2A, the color of the VSG box on the left and the colors of the VSG boxes in the contigs should be synchronized. They should either be all white (the color is perhaps not needed in this panel), or the one on the left should reflect the range of colors used.

2. “as LRRs are protein recognition motifs, this could be significant in gamete interactions.” This is highly unlikely because published data indicate that these specific proteins are highly enriched in the nucleus.

Reviewer #2: Main comments

The discussion of the sexual recombination between strains 1738 and J10 is somewhat brief, with several unanswered questions:

1) at which point in the Pseudotime analysis are the hybrid transcriptomes found? Can we reliably obtain a temporal order of events from these data, i.e., does gametocytogenesis precede epimastigote attachment to the epithelium?

2) Were the authors able to understand how meiotic recombination alters the distribution of SNPs throughout the chromosomes of the hybrids?

The expression analysis of Tb927.10.12080 is interesting and seems solid. The experimental design could impact interpretation, since a procyclin promoter is used to drive expression of the GFP reporter. We believe that from the description of the experiment in the materials and methods, the authors have images of midgut parasites showing that GFP is not expressed. These are an important control to illustrate that the regulation observed is indeed controlled by the 3´ UTR rather than the procyclin promoter and could be added to strengthen Figure 6.

Minor comments

Line 103 and legend figure S1. Is it 46 or 48 cells? The number differs at several places of the text. Please correct.

Line 117. “A further 62 1738 parasites” is a bit confusing. We would suggest using “strain 1738” or “strain J10” throughout the manuscript.

Line 130-140, the description of the number of transcriptomes was a little hard to follow. Could the authors inset a table into Fig 1A to help the reader?

Figure 1. The clusters at Fig. 1D are difficult to follow, especially since they are not presented in the same order as at Fig. 1C, which is a bit confusing. We suggest adding extra labels or keeping the same order.

Page 13: morphological features used for gamete identification were not very clear. Presumably, this is based on previous papers, but we could not find information in the manuscript.

Give a reference to the TrypTag paper for readers not specialised in trypanosomes (Halliday et al MBP2019).

Figure S3, should the x-axis label be “Genes detected”?

Line 239 “HMM” acronym not described

Line 290, no citation for Slingshot (though it appears in M&M)

The code used to analyze the data is not available at the stated URL.

Sebastian Hutchinson and Philippe Bastin, ESPCI & Institut Pasteur, Paris

Reviewer #3: There are a small Huber of typos, eg line 100 should be RBP6 not RBP 6

PLOS authors have the option to publish the peer review history of their article (what does this mean?). If published, this will include your full peer review and any attached files.

Reviewer #1: No

Reviewer #2: **Yes: **Sebastian Hutchinson and Philippe Bastin, ESPCI & Institut Pasteur, Paris

Reviewer #3: No

Figure Files:

Data Requirements:

Reproducibility:

References:

---

## [Editor Report · Decision Letter 1]

6 Feb 2022

Dear Dr Howick

We are pleased to inform you that your manuscript 'Single-cell transcriptomics reveals expression profiles of Trypanosoma brucei sexual stages' has been provisionally accepted for publication in PLOS Pathogens.

Best regards,

Lucy Glover

Guest Editor

PLOS Pathogens

David Horn

Section Editor

PLOS Pathogens

Kasturi Haldar

Editor-in-Chief

PLOS Pathogens

orcid.org/0000-0001-5065-158X

Michael Malim

Editor-in-Chief

PLOS Pathogens

orcid.org/0000-0002-7699-2064
---

## [Editor Report · Acceptance letter]

3 Mar 2022

Dear Howick,

We are delighted to inform you that your manuscript, "Single-cell transcriptomics reveals expression profiles of Trypanosoma brucei sexual stages," has been formally accepted for publication in PLOS Pathogens.

Best regards,

Kasturi Haldar

Editor-in-Chief

PLOS Pathogens

orcid.org/0000-0001-5065-158X

Michael Malim

Editor-in-Chief

PLOS Pathogens

orcid.org/0000-0002-7699-2064